# INSTANTSPLAMP: FAST AND GENERALIZABLE STENOGRAPHY FRAMEWORK FOR GENERATIVE GAUSSIAN SPLATTING

**Chenxin Li**[1]*, **Hengyu Liu**[1]*, **Zhiwen Fan**[2], **Wuyang Li**[1], **Yifan Liu**[1], **Panwang Pan**[3], **Yixuan Yuan**[1]†

[1]The Chinese University of Hong Kong,  [2]University of Texas at Austin,  [3]ByteDance

## ABSTRACT

With the rapid development of large generative models for 3D, especially the evolution from NeRF representations to more efficient Gaussian Splatting, the synthesis of 3D assets has become increasingly fast and efficient, enabling the large-scale publication and sharing of generated 3D objects. However, while existing methods can add watermarks or steganographic information to individual 3D assets, they often require time-consuming per-scene training and optimization, leading to watermarking overheads that can far exceed the time required for asset generation itself, making deployment impractical for generating large collections of 3D objects. To address this, we propose *InstantSplamp* (Instant Spltting Stamp), a framework that seamlessly integrates the 3D steganography pipeline into large 3D generative models without introducing explicit additional time costs. Guided by visual foundation models, *InstantSplamp* subtly injects hidden information like copyright tags during asset generation, enabling effective embedding and recovery of watermarks within generated 3D assets while preserving original visual quality. Experiments across various potential deployment scenarios demonstrate that *InstantSplamp* strikes an optimal balance between rendering quality and hiding fidelity, as well as between hiding performance and speed. Compared to existing per-scene optimization techniques for 3D assets, *InstantSplamp* reduces their watermarking training overheads that are multiples of generation time to nearly zero, paving the way for real-world deployment at scale. Project page: https://gaussian-stego.github.io/.

## 1 INTRODUCTION

Automatic 3D content generation has revolutionized diverse fields including gaming, virtual reality, and film production Samavati & Soryani (2023); Tang et al. (2023). Foundational techniques like image-to-3D and text-to-3D Poole et al. (2022); Tang et al. (2024); Liu et al. (2024) significantly reduce the manual labor required from professional 3D artists. These approaches simplify and democratize the creation process, enabling individuals without specialized expertise to contribute to 3D asset production. By making 3D design more accessible and efficient, these innovations foster an inclusive environment in the field, potentially reshaping the landscape of 3D content creation and distribution. Furthermore, this democratization opens up new possibilities for creative expression and innovation, as a wider range of perspectives and ideas can now be translated into 3D form, potentially leading to rich contents across various media platforms.

Research on 3D generation has evolved from Score Distillation Sampling (SDS) methods such as Poole et al. (2022); Lin et al. (2023); Liu et al. (2023a); Tang et al. (2023); Chen & Wang (2024), which optimize 3D representations to match the predictions of pre-trained 2D diffusion models, enabling the creation of

---

*Equal contribution

†Correspondence

**(a) 3D Generation with Existing Pre-scene 3D Watermarking Methods**

| Time for 3D Generation: **N*12 s** | Time for Watermarking: **N*6 min** |

User Prompt → 3D Gen (3D Large Generative Model) → Generated N Objects → Hidden Info. → Watermarked N Objects

**(b) 3D Generation with Our Generalizable 3D Watermarking (InstantSplamp)**

| Time for Our Unified 3D Generation & Watermarking: **N*12 s** |

User Prompt → 3D Gen → Generated & Watermarked N Objects ← Hidden Info.

*Ours slashes watermarking overhead from ~10x generation time to near-ZERO*

Figure 1: **Method Comparison of 3D Generation with Existing Per-scene Watermarking Methods vs. *InstantSplamp*.** (a) Traditional 3D generation requires separate steps for object generation and watermarking, leading to significant time overhead. (b) Our method, InstantSplamp, unifies 3D generation and watermarking into a single process, maintaining the generation time and reducing watermarking overhead to near-zero, significantly improving efficiency.

detailed 3D objects from text or single-view images. While SDS techniques have produced impressive results, they often struggle with challenges related to generation speed and diversity. Recent advancements have led to efficient feed-forward 3D-native techniques, trained on large-scale 3D datasets Deitke et al. (2023b;a), capable of generating 3D assets in just a few seconds. The latest research incorporates Gaussian splatting and optimized 3D backbones Tang et al. (2024), further enhancing texture details and geometric complexity. As we witness the rapid development of 3D asset generation, a new challenge arises: how to invisibly watermark the upcoming wave of generated 3D assets using steganography.

Traditional digital steganography methods have primarily focused on embedding hidden information within 2D images. However, the recent surge in generative AI and social media platforms has led to an explosion in the online sharing of generated content, driving the practical application of steganography in mainstream generative media. This trend has prompted research on embedding ownership information and metadata into generated content to ensure traceability, allowing users and content providers to protect their intellectual property. Additionally, efforts have been made to prevent content misuse by embedding covert backdoors, which prevent unauthorized re-creation of content through generative models Baluja (2017; 2019). These advancements address growing concerns over copyright protection and content misuse in the era of AI-generated media.

As 3D representation technologies powered by large generative models continue to evolve, we anticipate that generating and sharing large-scale 3D content will become as common as sharing 2D images and videos today. Existing watermarking techniques for 3D representations, such as those developed for meshes Ohbuchi et al. (2002); Praun et al. (1999), Neural Radiance Fields (NeRF) Li et al. (2023a), and Gaussian Splatting (GS) Kerbl et al. (2023), face significant time overhead due to the need for per-scene or per-object watermark training. While this process works for individual or small collections of assets, it becomes impractical for

large-scale 3D asset generation. For example, generating a 3D object might take only a few seconds, but watermarking each object can add several minutes, resulting in time costs tens of times greater than the generation itself. This inefficiency limits the scalability of current 3D watermarking methods. Motivated by this challenge, we aim to explore whether a scalable, generalizable 3D watermarking technique can be developed—one that eliminates the need for per-scene training and incurs no additional overhead, paving the way for efficient watermarking in large-scale 3D asset generation.

Driven by these inquiries, we propose *InstantSplamp*, an fast and generalizable 3D steganography framework that seamlessly integrates the 3D watermarking pipeline into large 3D generative models. Unlike existing steganography methods specialized for 3D content that require time-consuming per-scene optimization, *InstantSplamp* operates without introducing explicit additional time costs, making it practical for large-scale deployment. *InstantSplamp* utilizes visual foundation models to extract informative watermark embeddings, which are subtly injected into the intermediate features of a 3D Gaussian generation baseline through cross-attention mechanisms. For watermark recovery, a U-Net-based decoder retrieves the concealed information from images rendered at specific verification viewpoints. To balance rendering quality and information hiding, we introduce an adaptive gradient harmonization technique that aligns the gradients of the information hiding and rendering losses, optimizing both steganography and visual output. Extensive experiments demonstrate that *InstantSplamp* achieves a balance between rendering quality, hiding fidelity, and processing speed, reducing watermarking overhead—previously up to 30 times the generation time—to nearly zero. Our contributions are summarized as follows:

- We introduce InstantSplamp, a novel framework that seamlessly integrates fast and generalizable 3D steganography into large 3D generative models, enabling watermarking of 3D assets without additional time costs.

- We develop a unique approach that leverages visual foundation models to extract and inject watermark embeddings via cross-attention, allowing for effective embedding and recovery of hidden information in generated 3D Gaussian representations while preserving visual quality.

- We demonstrate the effectiveness of our framework across various deployment scenarios, showing significant improvements in efficiency by reducing watermarking overhead from approximately 30 times the generation time to nearly zero, thus enabling practical large-scale deployment.

- We empirically validate our framework's ability to embed and recover a wide range of signal modalities in 3D objects across different domains, achieving high recovery accuracy while maintaining rendering quality.

## 2 RELATED WORK

**Generative 3D Gaussian Splatting.** Gaussian splatting, introduced by Kerbl et al.Kerbl et al. (2023), is a powerful 3D representation known for its expressiveness and rendering efficiency. Enhancing details requires careful initialization and densificationChen et al. (2023b); Yi et al. (2023), but our research focuses on a feed-forward approach for autonomous 3D Gaussian generation. Unlike SDS-based methods, 3D-native feed-forward models trained on large datasets can generate 3D models in seconds Deitke et al. (2023b;a). While text-conditioned diffusion models for 3D formats have been explored Nichol et al. (2022); Jun & Nichol (2023); Liu et al. (2023c); Müller et al. (2023); Chen et al. (2023a), they often face scalability and quality issues. Recent advances include rapid NeRF prediction from single-view images Hong et al. (2023) and Instant3D Li et al. (2023b), combining text-to-multi-view diffusion and LRM for fast 3D generation. As 3D Gaussian Splatting evolves, it's timely to explore steganography tailored for this emerging field.

**Steganography for 2D Representation.** Deep learning has significantly advanced deep watermarking. Works like Hayes & Danezis (2017) and Zhu et al. (2018) introduced end-to-end learning paradigms where watermark encoders and decoders are refined via adversarial objectives, improving transmission fidelity and

robustness. Zeng et al. (2023) extended this by jointly optimizing a watermarked encoder and its detector using image datasets. More recent methods, such as Yu et al. (2022), integrate watermark encoding into the generative framework. The Stable Signature technique Fernandez et al. (2023) applies this in latent diffusion models by fine-tuning the latent decoder with a pre-trained watermark encoder. Similarly, Zhao et al. (2023) adapts this approach for unconditional diffusion models. In contrast, methods for language models, like Kirchenbauer et al. (2023), embed watermarks by modifying the output distribution without explicit training. While these advances were pivotal for traditional media, the rise of point-based Gaussian representations for 3D scenes calls for extending steganographic techniques from 2D to 3D, a critical area for future research as visual data representation evolves.

**Steganography for 3D Representation.** Within watermarking specialized for 3D contents, traditional approaches by Ohbuchi et al. Ohbuchi et al. (2002), Praun et al. Praun et al. (1999), and Wu et al. Wu et al. (2015) relied on Fourier or wavelet transformations applied to mesh structures. Recent innovations have expanded the field's scope: Hou et al. Hou et al. (2017) exploited 3D printing artifacts for watermarking, while Son et al. Son et al. (2017) and Hamidi et al. Hamidi et al. (2019) utilized mesh saliency to minimize vertex distortions and enhance robustness. Liu et al. Liu et al. (2019) explored watermarking for point clouds, focusing on vertex curvatures. A significant advancement came from Yoo et al. Yoo et al. (2022), who introduced a deep-learning method to embed messages in 3D meshes and extract them from 2D renderings. StegaNeRF Li et al. (2023a) further pioneered embedding messages into neural radiance fields (NeRF), enabling extraction of multimodal information from 2D renderings. A concurrent work, GS-Hider Zhang et al. (2024), shifts the focus of steganography from NeRF to 3DGS, leveraging its explicit representation and real-time rendering to embed hidden messages securely and invisibly without compromising rendering quality. However, current 3D watermarking methods typically require per-scene optimization, significantly increasing watermarking time and making them impractical for large-scale 3D generation scenarios.

# 3 METHOD

**Preliminary for Generative Gaussian Splatting.** Gaussian splatting Kerbl et al. (2023) represents 3D data using a collection of 3D Gaussians, where each Gaussian is defined by a center $\mathbf{x} \in \mathbb{R}^3$, a scaling factor $\mathbf{s} \in \mathbb{R}^3$, and a rotation quaternion $\mathbf{q} \in \mathbb{R}^4$. Additionally, each Gaussian has an opacity value $\alpha \in \mathbb{R}$ and a color feature $\mathbf{c} \in \mathbb{R}^C$, with spherical harmonics modeling view-dependent effects. These parameters, collectively denoted as $\mathbf{\Omega}$, define the $i$-th Gaussian as $\mathbf{\Omega}_i = \mathbf{x}_i, \mathbf{s}_i, \mathbf{q}_i, \alpha_i, \mathbf{c}_i$. Rendering projects 3D Gaussians onto the image plane and uses alpha composition in depth order to determine pixel color and opacity. Traditional 2D diffusion models generate images from a single viewpoint, lacking 3D viewpoint capabilities. To overcome this, recent methods fine-tune multiview diffusion models on 3D datasets, incorporating camera poses. This allows for multiview image generation from text or single images, generalizing to unseen objects. By integrating 3D spatial points and a consistency loss to ensure view coherence, these techniques improve performance and scalability when applied to 3D Gaussians.

**Pipeline Overview.** Conventional Image-to-3D generative models $F_{\mathbf{\Theta}}$ convert an input image $I$ into generative 3D Gaussians, parameterized as $\mathbf{\Omega} = F_{\mathbf{\Theta}}(I)$. The goal of our *InstantSplamp* is to discreetly embed proprietary steganographic information into the Gaussian generation process, causing imperceptible visual changes in the rendered output. Given hidden information $H$ to be embedded, the process involves two stages: embedding and decoding. During embedding, a hidden embedder $F_{\mathbf{\Phi}}$ integrates the features of the hidden image $f_H$ into the intermediate feature $f_I$ of the generation process. In the decoding stage, a checking view $P_c$ is selected from all available poses $P$. When rendering from this view using the generated Gaussians $\mathbf{\Omega}$, the embedded information $S$ is recovered via a decoder with learnable weights $\mathbf{\Psi}$. Fig.2 provides an overview of the *InstantSplamp* framework.

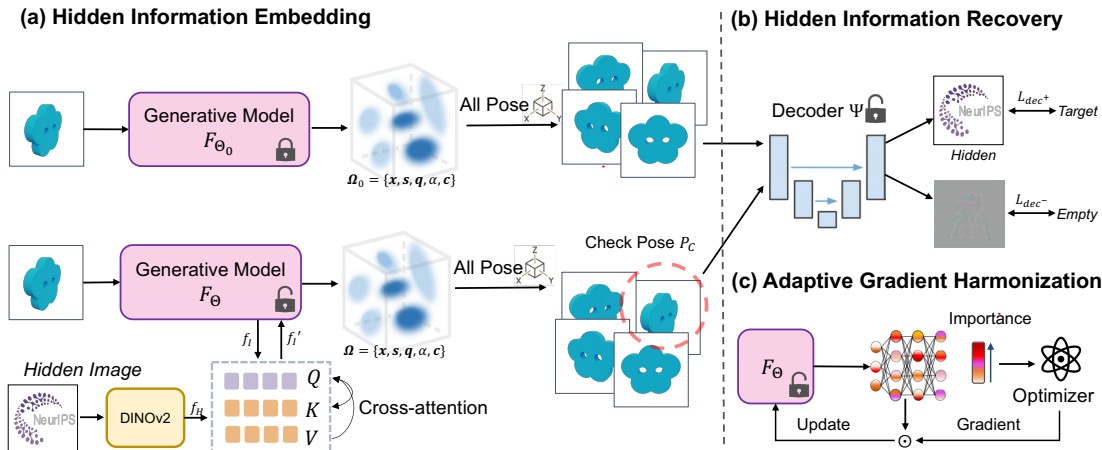

Figure 2: *InstantSplamp* training overview. During (a) Hidden Information Embedding, *InstantSplamp* incorporates the DINOv2 features of the hidden information into the intermediate feature of Gaussian generation via cross-attention. In (b) Hidden Information Recovery, a U-Net-based decoder is employed to retrieve the hidden information from the rendered image under the checking pose. Through the optimization process, (c) Adaptive Gradient Harmonization is used to maintain a balance between the rendering and hidden recovery.

## 3.1 HIDDEN INFORMATION EMBEDDING

Models that are operated under the demand to predict new views from a single input are inherently ill-posed, given that atypical 2D images can be projected from entirely distinct 3D representations. Consequently, when there is a requirement to embed 2D images into 3D representations, the anticipated 3D generative model is optimized to ensure that 2D images carrying hidden information can be inversely projected into the underlying 3D space in conformity with human perceptual understanding. Under such circumstances, directly embedding 2D watermark images into the feed-forward process of the equally ill-posed 3D generative model could potentially have deleterious effects on this inverse projection process. Inspired by the robust dense prediction capabilities exhibited by recent visual backbone models, particularly the semantic consistency demonstrated by spatial representations derived from the DINOv2 encoder compared to other encoders like CLIP Liu et al. (2023a;b); Radford et al. (2021), we advocate extrapolating the representational capabilities of the DINOv2 encoder to the to-be-encoded implicit watermark information. This effectively extracts the features of hidden information as $f_H$.

Although features derived from the visual base model provide a robust means of projecting the image into 3D space, this process can still result in the loss of detailed visual watermarks, thus impacting the visual quality of the hidden information. To address this challenge, we propose an early injection of the spatial details of the hidden information into the intermediate features derived from the image-to-3D Gaussian generation process. By integrating the image watermark into the intermediate feature $f_I$ of the generation process through cross-attention, the hidden image can effectively influence the update of the 3D representation via adaptive attention adjustment. Specially, the intermediate image feature $f_I$ can be updated by: $f_I := \text{Softmax}\left(\frac{K \cdot Q^T}{\sqrt{d}}\right) \cdot V$ (where $d$ denotes the length of the key and query features), with the key $K$, query $Q$ and value $V$ can be obtained by:

$$K = F_{\mathbf{\Phi}_K}(f_H), \quad V = F_{\mathbf{\Phi}_V}(f_H), \quad Q = F_{\mathbf{\Phi}_Q}(f_I) \tag{1}$$

where $F_{\mathbf{\Phi}_*}$ respectively for key $K$ and value $V$ represent the linear transformation layer applied to the feature $f_H$ derived from the hidden images, while that for query $Q$ represents the linear transformation layer applied

to the feature $f_I$. Therefore, for cross attention injection, we utilize the features of image watermarks to weight 3D features through cross attention. Effectively, the strategy promotes the propagation of visual cues from image watermarks through network propagation Li et al. (2024); Li et al.; Zhang et al. (2021).

## 3.2 HIDDEN INFORMATION RECOVERY

Given the set of camera poses $P$ for the image rendered with the generated 3D Gaussians, we aim to extract the concealed information $H$ when rendering at a specific checking viewpoint $P_C$ in the image $\mathbf{\Omega}(P_C)$. It is crucial to prevent the emergence of false positives of hidden watermarkers in the renderings of Gaussians, generated at checking viewpoint by the original generative network $F_{\mathbf{\Theta}_0}$, as $\mathbf{\Omega}_0(P_C)$, which lacks steganographic features. Even if the images rendered by $\mathbf{\Omega}_0$ and $\mathbf{\Omega}$ visually appear identical, we aim to minimize the following contrastive loss term:

$$\mathcal{L}_{dec^+} = |F_{\mathbf{\Psi}}(\mathbf{\Omega}(P_C)) - H|, \quad \mathcal{L}_{dec^-} = |F_{\mathbf{\Psi}}(\mathbf{\Omega}_0(P_C)) - \varnothing|, \tag{2}$$

where $\varnothing$ is a meaningless constant image that can be pre-defined by the users. Effectively, $L_{dec^+}$ serves as a regularization term, aiding the decoder in recovering the embedded image patterns based on the rendering obtained from the model. On the contrary, $L_{dec^-}$ prevents the decoder from erroneously generating any seemingly reasonable image patterns when rendering is given from the standard generated Gaussians without any hidden signal. The decoder $\mathbf{\Psi}$ can be conveniently implemented as U-Net to decode $H$ into the form of a 2D image. Although the above discussion is primarily focused on hiding images, our framework can easily be extended to embed other modalities such as strings, text, and even audio, all of which can be represented as 1D vectors. We can simply modify the architecture of $\mathbf{\Psi}$ to have a 1D prediction branch.

## 3.3 PRESERVING PERCEPTUAL IDENTITY

**Overall Loss.** We retain the standard photometric error in steganography learning to maintain the Gaussian rendering fidelity across any views between the steganographic one and the original one: $\mathcal{L}_{rgb} = |\mathbf{\Omega}(P) - \mathbf{\Omega}_0(P)|$. The overall training loss of the framework can be formulated as follows, given the input reference image $I$ and hidden image $H$:

$$\mathcal{L}(I, H; \mathbf{\Theta}, \mathbf{\Phi}, \mathbf{\Psi}) = \lambda_1 \mathcal{L}_{dec^+} + \lambda_2 \mathcal{L}_{dec^-} + \lambda_3 \mathcal{L}_{rgb}. \tag{3}$$

where $\lambda_1, \lambda_2, \lambda_3$ is the trade-off coefficients. The parameters that are optimized by the above loss is listed as the generative model $\mathbf{\Theta}$, the hidden embedder $\mathbf{\Phi}$ and hidden decoder $\mathbf{\Psi}$. Please note that while the parameters of the Gaussian $\mathbf{\Omega}$ are present in the loss formula, during the optimization of these parameters, we directly update the generative network $\mathbf{\Theta}$, thereby indirectly influencing the generated Gaussian by $\mathbf{\Omega} = F_{\mathbf{\Theta}}(I)$,

**Adaptive Gradient Harmonisation.** Given our objective to embed information without altering the visual perception of the rendered output, one might intuitively consider penalizing the deviation between $\mathbf{\Theta}$ and $\mathbf{\Theta}_0$ as an effective regularization. However, we have found that naively incorporating penalties for deviations of all weights impedes the generative network's ability to alter its weights for steganographic purposes. Instead, we take into account two key insights: 1) The quality of the rendering and the hidden information both exert influence on the generative network, leading to potentially conflicting demands as the network must conceal hints about the hidden information in certain aspects of the rendering. 2) The weights of the generative network do not equally contribute to the quality of the GS rendering and exhibit strong sparsity. Inspired by these insights, in what follows, we introduce an adaptive gradient harmonization strategy to embed hidden information into specific weight groups of the generative network, where the gradient update requirements for rendering and hidden information are aligned.

Formally, given the weights $\mathbf{\Theta} \in \mathbb{R}^N$, we compute a gradient mask $\mathcal{M} \in \mathbb{R}^N$ that indicates whether the gradients with respect to rendering and hidden embedding are harmonious across all weights:

$$\mathcal{M} = \mathbb{I}\left( \text{Cos}\left( \frac{\partial \mathcal{L}_{rgb}}{\partial \mathbf{\Theta}}, \frac{\partial (\mathcal{L}_{dec^+} + \mathcal{L}_{dec^-})}{\partial \mathbf{\Theta}} \right) > 0 \right) \tag{4}$$

where $\mathrm{Cos}(\cdot, \cdot)$ denotes the cosine similarity between the two gradient components, and $\mathbb{I}$ is an indicator function that is equal to 1 if the condition is true and 0 otherwise. Notably, the cosine similarity here is computed globally to determine the overall alignment of gradients, acting as a global switch for the entire set of weights. We apply the mask to the gradient as $\frac{\partial \mathcal{L}}{\partial \Theta} \odot \mathcal{M}$ when optimizing $\Theta$ based on the total loss $\mathcal{L}$, where $\odot$ represents the Hadamard product. By doing so, this masking mechanism ensures that either all gradients are retained or all are discarded based on their global alignment, thereby simplifying the harmonization process. Effectively, the gradients related to information embedding, which are inconsistent with the objective of maintaining the rendering quality from the pre-trained generative model that tends to generate high-quality GS representations, are "masked out" on those conflicting weights.

## 4 EXPERIMENT

### 4.1 EXPERIMENTAL SETUP

**Dataset and Evaluation** We train our model on a filtered subset of the Objaverse dataset Deitke et al. (2023c), excluding low-quality 3D models such as partial scans and those without textures. This filtering process results in a final collection of approximately 80K 3D objects. For training, 100 objects are randomly selected, while a separate test set of 100 unseen objects is reserved for evaluation. We render RGBA images from 40 camera views at a resolution of $256 \times 256$ for both training and testing. To evaluate the quality of the recovered hidden information, we assess PSNR, SSIM, and LPIPS. All metrics are calculated on the test set and averaged across all scenarios and embedded images Chen et al. (2022); Li et al. (2022a); Liang et al. (2021). For subjective evaluation, 360-degree rotational videos of 3D Gaussians generated by different methods are rendered for a collection of 30 images. Each of the 30 samples, randomly selected from different methods, is presented to 20 volunteers who are asked to score based on overall visual quality.

**Implementation.** Our approach is deployed over an advanced image-to-3DGS model, LGM Tang et al. (2024), which serves as the Gaussian generator. A simple U-Net is utilized as the decoder for hidden information. Upon the Gaussian generation foundation, we fine-tune a LoRA Hu et al. (2021) for each watermark image. This training routine typically necessitates training over N=100 objects for approximately 30 epochs, which takes around 20 minutes, and subsequently, it can be generalized to other unseen objects. We employ the AdamW optimizer for optimization, with a learning rate of $1e-4$. For hyper-parameters in Eq. (3), we set the weight $\lambda_1 = 0.3, \lambda_2 = 1, \lambda_3 = 0.1$ for all experiments.

Table 1: Quantitative comparison in rendering and hidden information recovery.

| Method | Rendering | | | | Hidden Recovery | | | |
|---|---|---|---|---|---|---|---|---|
| | PSNR↑ | SSIM↑ | LPIPS↓ | Subj.↑ | PSNR↑ | SSIM↑ | LPIPS↓ | Subj.↑ |
| Init. Render | 20.48 | 0.8522 | 0.1181 | 5.00 | N/A | N/A | N/A | N/A |
| LSB Chang et al. (2003) | 20.45 | 0.8518 | 0.1185 | 3.95 | 8.36 | 0.2091 | 0.5379 | 1.20 |
| DeepStega Baluja (2017) | 20.43 | 0.8513 | 0.1197 | 3.21 | 12.11 | 0.2847 | 0.4432 | 1.83 |
| StegaNeRF Li et al. (2023a) | 18.53 | 0.8362 | 0.1756 | 2.30 | 31.87 | 0.9659 | 0.0114 | 3.25 |
| *InstantSplamp* (Ours) | 20.45 | 0.8519 | 0.1189 | 4.11 | 32.97 | 0.9808 | 0.0082 | 3.67 |

### 4.2 EMBEDDING 2D VISUAL CONTENTS AS HIDDEN INFORMATION

**Baseline Models.** Since no prior work exists on steganography for Gaussian Splatting, we establish baselines from 2D image steganography by fine-tuning large generative models with watermarked images. We implement two methods: a traditional approach, Least Significant Bit (LSB Chang et al. (2003)), and a deep learning pipeline, DeepStega Baluja (2017)). Additionally, we extend StegaNeRF Li et al. (2023a) by

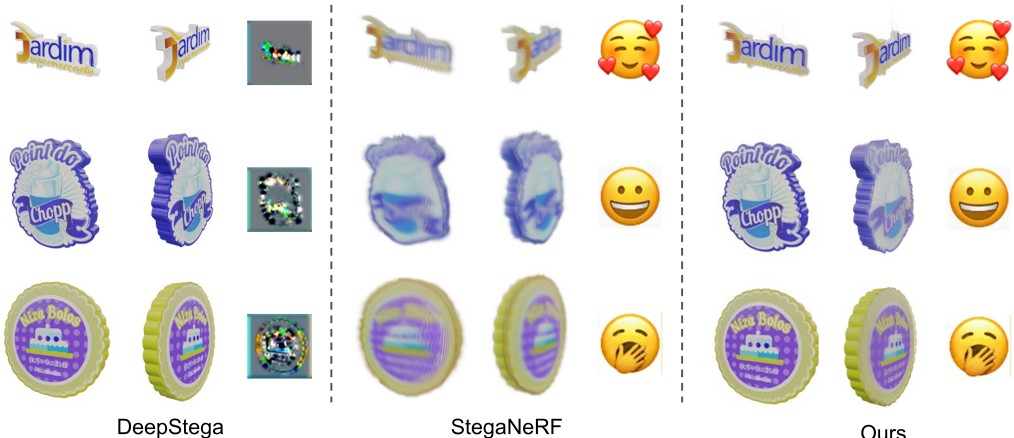

Figure 3: Qualitative comparison on the test objects of the Objaverse dataset. Within each column, we show the rendering images on check pose and and the recovered hidden images.

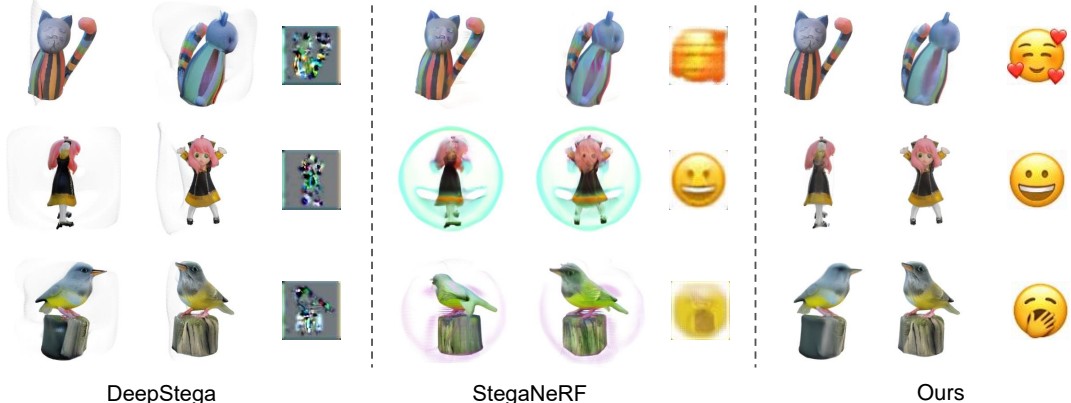

Figure 4: Quantitative comparison on widely-used test images by image-to-3D models. Within each column, we show the rendering and the recovered hidden images.

redesigning its framework to a Gaussian Splatting-based pipeline for embedding recoverable watermarks, as a watermarking baseline for Gaussian Splatting. These baselines provide essential benchmarks and insights into the unique challenges of information hiding in Gaussian Splatting.

Tab.1 presents quantitative results from testing on selected objects in the Objaverse dataset. Despite challenges in recovering embedded information through 2D steganography methods and maintaining rendering quality with StegaNeRF, our model minimizes the impact on rendering quality, as evidenced by PSNR scores. Fig.3 compares steganography on three objects from the Objaverse dataset, showcasing our model's ability to retain the original rendering details while precisely recovering the watermark. These results highlight the superiority of our approach in balancing information hiding capacity and visual fidelity, a critical factor in practical 3D steganography applications.

Our framework has been rigorously tested in real-world image-to-3D deployment scenarios. As shown in Fig.4, we applied our method to embed watermarks in various prevalent use-cases and compared its performance to other state-of-the-arts Ye et al. (2023); Wang et al. (2022); Li et al. (2022b; 2021b) to assess the generalizability of *InstantSplamp*. Compared to alternative techniques, *InstantSplamp* consistently preserves the details of the original, unwatermarked renderings while accurately extracting the watermark, even for unseen objects Wang et al. (2022); Li et al. (2021a); Zhang et al. (2021); Xu et al. (2022); Ding et al. (2022). This demonstrates the applicability of our method in practical 3D asset production environments. Moreover, the model's ability to generalize to unseen objects underscores its potential for widespread adoption across industries, like entertainment and design, where protecting intellectual property in 3D assets is crucial.

## 4.3 Embedding Multimodal Contents as Hidden Information

We further explore the capabilities of *InstantSplamp* in integrating multimodal hidden information such as text, QR codes, audio, and video into generated objects. Compared to traditional image watermarking techniques, utilizing multimodal data provides a richer and more comprehensive information set. However, it also introduces the challenge of managing an increased volume of embedded information. To address this, we enhance the decoder network by implementing a modality-specific decoder for each type of input modality. This allows for more effective recovery and utilization of the embedded information. Figure 5 illustrates the successful recovery of multimodal embedded signals from three generated objects, demonstrating the effectiveness of our approach. We use different metrics to evaluate the quality of recovery for different modalities: ACC for text embedding recovery, SSIM for QR code watermark recovery, and PSNR for audio recovery. Our results indicate that these metrics are suitable for assessing the quality of recovered information across various modalities.

Furthermore, we experiment with embedding a 16-frame video into assets represented by Gaussian representations. The success of this experiment suggests that the *InstantSplamp* framework can be easily adapted to accommodate multimodal information with high recovery performance, all while maintaining the rendering quality of the generated objects. This flexibility and efficiency in handling multimodal information open up numerous potential applications for watermarking in generative 3D Gaussian contexts. For instance, this could be particularly useful in digital rights management, where multimodal watermarks could provide additional layers of security and authentication. Additionally, in fields like entertainment or education, multimodal embedding could enable the creation of interactive and enriched content by seamlessly integrating various types of media into a single 3D object.

## 4.4 Ablation Studies

**Effectiveness of Components** In Tab. 2, we present an analysis of the effect of removing each component of *InstantSplamp*. This analysis is crucial in understanding the contribution of each component to the overall performance of the system. The variant termed as *No All Components* discards all the proposed additions to the system. This version only retains the fundamental rendering and steganographic loss, providing a baseline for comparison. On the other hand, the variant *No DINOv2* eliminates the newly introduced DINOv2. This component is used as the feature extractor for the hidden image. Instead of DINOv2, this variant employs the CLIP Image encoder, providing a different approach to feature extraction. The variants *No Cross Attention* and *No Gradient Harmoni.* respectively eliminate the cross-attention of hidden information on the intermediate features of Gaussian generation and the newly introduced gradient harmonization strategy. Through this analysis, it becomes clear that when any component is removed, the performance drops accordingly, underscoring the effectiveness of our design in each component.

**Robustness Analysis** As shown in Fig.6, we can observe that *InstantSplamp* is robust against common perturbations, such as JPEG compression and Gaussian noise. We provide the SSIM of rendered views (blue) and recovered hidden images (green). The curves are representative of mean accuracies that have been

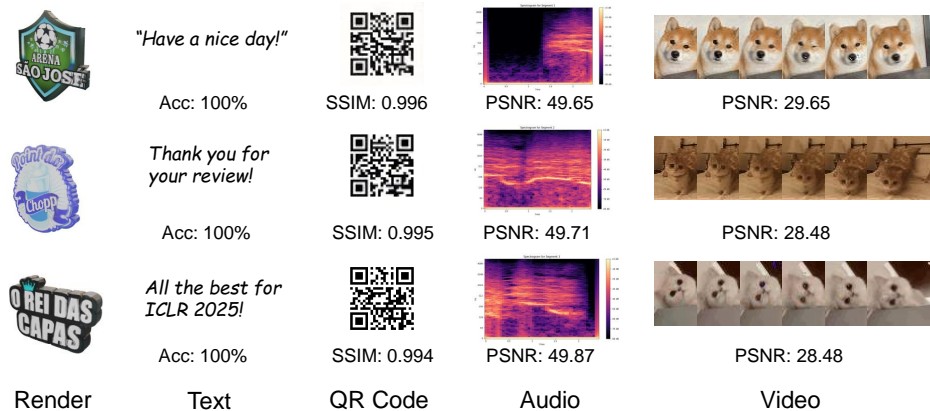

Figure 5: Quantitative results of *InstantSplamp* with multimodal information being embedded.

Table 2: Ablation study on the proposed key components of *InstantSplamp*.

| Method | Rendering | | Hidden Recovery | |
|---|---|---|---|---|
| | PSNR↑ | SSIM↑ | PSNR↑ | SSIM↑ |
| No All Components | 18.39 | 0.7904 | 27.32 | 0.8870 |
| No DINOv2 | 20.04 | 0.8507 | 30.60 | 0.9695 |
| No Cross Attention | 19.66 | 0.8395 | 30.15 | 0.9669 |
| No Gradient Harmoni. | 20.11 | 0.8529 | 31.17 | 0.9629 |
| Full Model (Ours) | 20.45 | 0.8519 | 32.97 | 0.9808 |

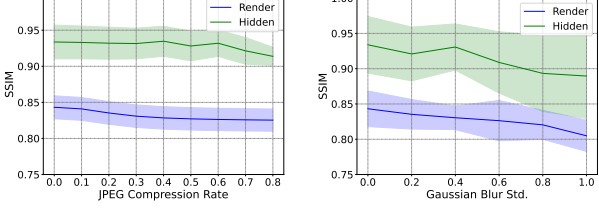

Figure 6: Robustness Analysis over (a) JPEG compression and (b) Gaussian blur.

computed across selected scenes. The shaded regions, on the other hand, signify a range of 0.5 standard deviation. This suggests a certain degree of variability within the data even with the limited data quality Pan et al. (2023). The results of this analysis strongly suggest that the ability of the *InstantSplamp* to recover hidden information remains consistent and resilient, even when exposed to a wide variety of JPEG compression levels and deteriorations caused by Gaussian blur. This robustness extends the practical applicability of *InstantSplamp* to real-world scenarios where image manipulations and transformations are common, ensuring the integrity of embedded information across various digital environments and transmission channels.

## 5 CONCLUSION

This paper addresses the challenge of integrating steganographic information into 3D content generation, focusing on emerging techniques like Gaussian Splatting. We introduce *InstantSplamp* (Instant Splitting Stamp), a novel framework that seamlessly embeds copyright or proprietary information into the generation process of 3D assets while maintaining original visual quality. Unlike existing methods that require time-consuming per-scene watermark optimization, *InstantSplamp* operates without introducing additional time costs, making it practical for large-scale deployment. Our approach leverages visual foundation models to subtly inject hidden information during asset generation, enabling effective watermark embedding and recovery across various 3D assets. *InstantSplamp* has been extensively tested across various scenarios, showcasing its capability to embed diverse hidden information while optimizing rendering quality, hiding fidelity, and processing efficiency. By significantly reducing watermarking overhead, *InstantSplamp* enables generalizable, scalable and usable applications in the growing field of 3D asset generation and protection.

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
