# OpenReview forum: "InstantSplamp: Fast and Generalizable Stenography Framework for Generative Gaussian Splatting"
_ICLR.cc/2025/Conference — ICLR 2025 Poster_

### Official Review · Reviewer_ec5a · 2024-10-29

**Soundness:** 3
**Presentation:** 3
**Contribution:** 3
**Rating:** 8
**Confidence:** 5

**Summary:**

The author skillfully integrates the visual foundation models into 3D steganography by leveraging cross-attention mechanisms to embed hidden information during the generation process. The proposed framework optimizes the balance between rendering quality and watermark fidelity, ensuring minimal distortion while preserving the integrity of the embedded data. The author validates the practicality of this approach through extensive experiments on various 3D assets.

**Strengths:**

1. The proposed InstantSplamp framework is highly innovative as it integrates watermarking directly into the 3D generation process, significantly reducing time overhead, making it practical for large-scale deployment.
2. The methodology leverages visual foundation models and cross-attention mechanisms in a novel way to embed and recover hidden information effectively, while maintaining high rendering quality.
3. The paper presents strong empirical validation across multiple deployment scenarios, demonstrating the method’s efficiency and generalizability with various 3D objects and modalities, including images, text, QR codes, and even video.
4. The use of adaptive gradient harmonization to balance rendering fidelity and information hiding represents a practical and insightful solution to a common challenge in steganography, ensuring minimal visual quality degradation.

**Weaknesses:**

1. While Figure 1 illustrates the time efficiency improvements of the proposed method for watermarking, could you provide some quantitative experimental results to further emphasize this point?
2. The robustness testing only considers two types of corruptions (JPEG compression and Gaussian blur), which seems limited in scope. It would be valuable to include additional forms of corruption, such as noise, scaling, or cropping, for a more comprehensive evaluation. Additionally, a comparative robustness analysis with other state-of-the-art methods is missing, which would provide a clearer understanding of how InstantSplamp performs under various conditions.
3. How does the proposed method compare with other 3D watermarking approaches targeting binary messages, such as those for NeRF or other 3D representations? Specifically, it would be helpful to see a comparison of performance in embedding and recovering complex information, as well as any advantages InstantSplamp may have over these existing methods.

**Questions:**

See problems mentioned in Weaknesses.

---

### Official Review · Reviewer_BP2h · 2024-10-31

**Soundness:** 2
**Presentation:** 3
**Contribution:** 2
**Rating:** 3
**Confidence:** 5

**Summary:**

The paper introduces an end-to-end framework for 3DGS steganography, embedding an image during Gaussian generation and recovering it from a specific rendering via a decoder network. In the hiding stage, the framework employs a cross-attention mechanism to seamlessly integrate the hidden image features into the spatial details of the intermediate Gaussian features. In the recovery stage, the decoder network extracts the hidden image exclusively from the rendering of a specific viewpoint. Additionally, an adaptive gradient harmonization technique is introduced, which functions as a masking mechanism, embedding the hidden information within certain model weights to preserve both steganographic ability and the visual quality of the renderings.

**Strengths:**

- The paper proposes a generalizable steganography mechanism that avoids additional time costs and modifications to the original Gaussian generation process.
- The experimental results in the paper demonstrate that the steganography capability of 3DGS surpasses that of similar methods applied in NeRFs.

**Weaknesses:**

- The method is similar to StegaNeRF and lacks sufficient novelty.
- The experimental baselines are too limited. Notably, an existing method, GS-Hider: Hiding Messages into 3D Gaussian Splatting, already achieves multi-scene information hiding within a 3DGS model.
- The experiments lack an analysis of steganographic capability, such as different capacity, resistance against steganalysis networks and robustness to additional distortions.

**Questions:**

- Does the proposed method exhibit superior steganographic capability compared to existing 3DGS steganography techniques?
- Is it possible to increase the capacity for embedding additional images within the steganographic framework?

---

### Official Review · Reviewer_mvPP · 2024-11-01

**Soundness:** 3
**Presentation:** 1
**Contribution:** 2
**Rating:** 6
**Confidence:** 3

**Summary:**

This paper presents a method, named InstantSplamp, to insert watermark information in generated 3D contents. The method is generalized, which means it does not need per-scene optimization. The time cost is extremely faster than previous methods.

**Strengths:**

1. The result achieved by the proposed method is much better than previous methods, especially for the hidden recovery performance.
2. The method does not need per-scene optimization, which is a generalized model and thus leads to faster speed.

**Weaknesses:**

1. The notations are not clear, it is hard to understand the figure 2 without notations.
2. It's hard to understand the "AdaptiveGradientHarmonisation", the cosine similarity in Eq. (4) seems to be calculated based on all parameters. In this way, the similarity is not a vector value, so what does the mask stand for?
3. The training is only conducted on one model, i.e., LGM. This limits the application. Authors should show more results on different 3D generative models with different representations.

**Questions:**

See weakness

---

> ### Comment · Reviewer_mvPP · 2024-11-26
>
> Thanks for the authors’ clarification and I am willing to raise my score.

---

### Official Review · Reviewer_BDqe · 2024-11-04

**Soundness:** 3
**Presentation:** 2
**Contribution:** 2
**Rating:** 5
**Confidence:** 4

**Summary:**

InstantSplamp introduces a fast, scalable framework that embeds hidden information like copyright tags into 3D generative models without additional processing time. Leveraging visual foundation models and cross-attention mechanisms, this approach integrates watermarking directly within the 3D Gaussian Splatting process. Unlike traditional methods requiring per-scene optimization, InstantSplamp minimizes overhead to nearly zero, enabling efficient large-scale deployment of watermarked 3D assets. A U-Net-based decoder recovers the hidden information, balancing visual quality with steganographic fidelity. This innovation addresses scalability challenges in 3D asset generation and protection, optimizing both watermark embedding and retrieval.

**Strengths:**

- It indeed doesn’t require per-scene optimization, which gives it a time advantage.

- The idea of injecting a watermark directly into the 3D generation model is good.

**Weaknesses:**

- In Figures 3 and 4, the 3D assets generated by your method show some artifacts in rendering, and the colors are somewhat distorted. Injecting the watermark affects the visual quality. Although it performs much better compared to StegaNeRF, the impact on visual quality due to watermark injection seems counterproductive.
- There is no 360-degree visual quality demo, and only two views are provided, which makes it hard to assess the rendering quality of the 3D assets and the quality of watermark extraction. It’s unclear whether the rendering quality of the 3D assets is 3D consistent.
- From the data in Table 1, the rendering quality of your method is not significantly better than LSB or DeepStega, and there’s no comparison with the latest method, GS-Hider.

GS-Hider: Hiding Messages into 3D Gaussian Splatting

**Questions:**

See Weaknesses.

---

### Meta-Review · Area_Chair_zshd · 2024-12-19

**Metareview:**

InstantSplamp introduces a scalable method to embed watermarks into 3D generative models directly within the 3D Gaussian Splatting process, eliminating per-scene optimization and significantly reducing time costs while maintaining reasonable visual quality and reliable watermark recovery.

Several key strengths were highlighted by multiple reviewers: (1) the generalizable network, which eliminates the need for per-scene optimization and significantly enhances efficiency; (2) the direct embedding of watermarks into the model, enabling large-scale training and deployment; (3) the method's effectiveness validated by the SOTA performance.

The paper has several weaknesses noted by multiple reviewers. First, the rendering quality is concerned, and results with limited views are validated. Second, reviewers are concerned about the comparison with GS-Hider, which can be considered concurrent. Additionally, the scope of robustness testing is limited, and the method's training is restricted to a single model (LGM), raising questions about its generalizability. As verified by the ACs, the authors addressed most of the issues in their rebuttal.

Given the approach's effectiveness, the clarification of concerns, and the mostly borderline to positive ratings (2 negative cases are discussed in the following), the ACs are inclined to accept the paper.

**Additional Comments On Reviewer Discussion:**

The paper received a `5: marginally below the acceptance threshold` from reviewer `BDqe`. The reviewer's major concern lies in the suspicion of the effectiveness of embedding hidden information, and whether the decoder memorizes the watermark information. Although the reviewer did not respond to the final clarifications, the authors addressed this issue by demonstrating, through comparisons of freeze/unfreeze generator and decoder settings, that memorization is not the cause.

The paper received a `3: reject, not good enough` from the reviewer `BP2h`, who raised concerns about the lack of methodological differences, the absence of comparisons with GS-Hider, and insufficient analysis of steganographic capabilities. While the authors addressed these issues, the reviewer's response appears to lack relevance to the clarifications provided. As a result, the weight of this rating has been reduced.

---

### Decision · Program_Chairs · 2025-01-22

Accept (Poster)